Between darkness and light: spring habitats provide new perspectives for modern researchers on groundwater biology

http://orcid.org/0000-0001-6071-8194 Manenti Raoul 1 2 raoul.manenti@unimi.it
Piazza Beatrice 1
1 Department of Environmental Science and Policy, Università degli Studi di Milano , Milano, Lombardia , Italy
2 Parco Regionale del Monte Barro, Laboratorio di Biologia Sotterranea “Enrico Pezzoli” , Galbiate , Italy
Jaffé Rodolfo
Electronic publication date: 2021 Jul 26
Publication date: 2021
Volume: 9
Electronic Location ID: e11711
Received 2021 Apr 13; Accepted 2021 Jun 10
Copyright: © 2021 Manenti and Piazza
Copyright year: 2021
Copyright holder: Manenti and Piazza
License: This is an open access article distributed under the terms of the Creative Commons Attribution License, which permits unrestricted use, distribution, reproduction and adaptation in any medium and for any purpose provided that it is properly attributed. For attribution, the original author(s), title, publication source (PeerJ) and either DOI or URL of the article must be cited.
License URL: https://creativecommons.org/licenses/by/4.0/

Keywords: Seepage, Source, Shallow subterranean habitats, Adaptation, Subterranean biology, Cave

Funding: University of Milan APC The University of Milan supported the publication fee through the APC initiative. The funders had no role in study design, data collection and analysis, decision to publish, or preparation of the manuscript.

==============================
Springs are interfaces between groundwater and surface habitats and may play an important role in the study of subterranean animals. In this systematic evidence review and meta-analysis, we explore whether observations of stygobionts in springs are relevant and more common than observations of epigean animals in groundwater. We searched the Web of Science database for papers on groundwater fauna and spring fauna. For each paper we found, we recorded whether the paper reported the occurrence of typical stygobionts in springs, of surface animals in groundwater, or of the same taxa in both habitats. If so, we recorded how many such species were reported. We also recorded the scientific discipline of each study and the year of publication. Our search yielded 342 papers. A considerable number of these papers reported stygobionts in springs: 20% of papers dealing with groundwater fauna and 16% of papers dealing with spring fauna reported the occurrence of stygobionts in spring habitats. Both the number of papers that mentioned stygobionts in springs, and the number of stygobiont species that were documented in springs, were higher than equivalent measures for the occurrence of surface fauna underground. We also detected a positive relationship between year of publication and the number of reports of stygofauna in springs. To broaden the insights from biological research on underground environments, we suggest that springs should be considered not only as simple sampling points of stygobionts but also as core stygobiont habitats.

Introduction

The zoologist Botosaneanu (1998) defined springs as the “doors on River Styx,” the river of the Greek mythological underworld. Other biologists who study subterranean environments and groundwaters similarly consider springs to be openings that allow them to see the inhabitants of an otherwise inaccessible environment (Culver, Holsinger & Feller, 2012; Fiasca et al., 2014; Galassi et al., 2014; Pipan & Culver, 2012; Pipan et al., 2012). This view of spring habitats as windows into a different environment is particularly true in non-karst areas, where the lack of caves prevents human exploration of the subterranean realm and springs are often the only way to access groundwater organisms (Manenti & Pezzoli, 2019). Springs are interfaces between groundwater and surface freshwaters with a great variety of connectivity, permeability and biodiversity (Gibert, Mathieu & Fournier, 1997; Kresic, 2010; Springer & Stevens, 2009). The interplay of both subterranean and epigean habitat features that characterises springs (Alfaro & Wallace, 1994; Cantonati, Gerecke & Bertuzzi, 2006, makes challenging to define the spatial boundaries of these ecotones (Gibert, Mathieu & Fournier, 1997).

In some cases the transition between surface and subterranean habitats, may also be abrupt; indeed, the magnitude of this transition strongly depends on the morphology of the spring and can be mutable with daylight. Some springs represent an abrupt shift from the subterranean environment to the surface, whereas others, like the natural emitting caves (such as caves from which subterranean streams flow outside) of artificial draining galleries (such as galleries built to collect groundwater), represent extended ecotonal environments (Balland, 1992; White, 2019). The border between the subterranean and surface environment can be particularly distinct during daytime, when it is strictly demarcated by the sun. Aside from sunlight, the differences that distinguish subterranean and surface environments in a spring, even across the few meters or centimetres that may characterize a spring with a sudden interface (Fig. 1), include the availability of trophic resources, the density of potential predators, and microclimate conditions (Barzaghi et al., 2017; MacAvoy et al., 2016; Manenti, Siesa & Ficetola, 2013; Von Fumetti & Nagel, 2011).

Figure 1 Diagram of a spring showing differences between surface and groundwater habitats during day (A) and night (B).

During night the border between surface waters and groundwaters softens and stygobionts can move outside interacting with surface invertebrates and vertebrates including predators (here represented by a diurnal predator fish and a nocturnal predator salamander larva). White silhouettes represent stygobionts, black silhouettes represent potential predators (fish and salamanders), and brown silhouettes surface aquatic invertebrates. The drawing is modified from Andrea Melotto and Benedetta Barzaghi.

Because springs are border habitats, it can be difficult for biological studies to consider springs in their entirety; this difficulty has limited the potential for insights from springs to drive stronger advances in different fields of research. For example, studies that focus on springs often only consider a surface perspective and neglect the role played by groundwater (Manenti & Pezzoli, 2019), whereas in karst areas, scientists studying the subterranean environment see springs as “access points” that can be used to sample the groundwater fauna living in different subterranean, underwater environments, such as the phreatic zone of karst aquifers (Malard et al., 2002). This latter view reflects the scarce consideration that is often given to springs and may limit a more general understanding of the ecological role of border habitats. As some studies have already suggested, transition zones are important for regulating ecosystem processes and the flow of organic matterand organisms between surface (epigean) and underground (hypogean) habitats (Moseley, 2009; Plenet & Gibert, 1995; Prous, Ferreira & Martins, 2004). With this opinion paper based on a systematic review of the recent scientific literature, we aim to stimulate a change in the conception of and in the approach to springs by studies dealing with stygobionts and groundwater fauna. Particularly we want to underline that springs have the potential to reveal general patterns related to the zoology of stygobionts.

Stygobionts are obligate groundwater-dwellers; the etymology of the word “stygobiont” reflects the fact that these species, and stygofauna more broadly, are “of the River Styx.” These organisms have evolved adaptations specific to the underground freshwater habitats in which they spend their entire life cycle (Trajano & De Carvalho, 2017). Stygobionts often exhibit morphological features associated with their underground habitat. These characteristics, such as blindness and depigmentation, are commonly referred to as troglomorphisms (Pipan & Culver, 2012; Romero, 2009), and they limit stygobionts’ ability to exploit surface environments. However, at night, the constraints are generally less clear and surface borders (i.e., springs) may become more permeable by stygofauna; for example, some samplings of springs’ fauna have reported during night the occurrence of organisms considered to be strict stygobionts (Bressi, Aljancic & Lapini, 1999; Fišer, 2019; Manenti & Barzaghi, 2020; Manenti & Barzaghi, 2021), as it has been observed also for the twilight zones of terrestrial caves (Mammola & Isaia, 2018). One such case is that of Stygobromus amphipods, which are believed to regularly leave hypotelminorheic habitats to feed (Culver, Pipan & Gottstein, 2006; Culver & Pipan, 2014). Nevertheless, these findings are often viewed as exceptions or accidental events, and the use of springs is rarely mentioned as a trait of stygobiont biology. Observations of surface animals in caves have been similarly overlooked in the past (Sket, 2008) and improperly seen as accidental; such “accidental” observations have recently been described for both groundwater and terrestrial subterranean habitats (Ficetola et al., 2018; Lunghi, Manenti & Ficetola, 2014a; Manenti, 2014).

Stygobionts are the main focus of subterranean biology and are usually studied using two distinct approaches. The first approach includes intense taxonomic investigations focused on the discovery and description of new taxa. The second approach views caves as powerful natural laboratories for evolutionary, ecological and behavioural studies on their inhabitants (Culver & Pipan, 2014; Culver & Pipan, 2019). The idea of caves as natural laboratories, first postulated by the speleologist Édouard-Alfred Martel (1894), has been espoused for more than one hundred years of subterranean studies (Poulson & White, 1969) and recently updated and broadened in the current context of biological research (Mammola, 2019; Mammola et al., 2020) . However, of the relatively large number of caves that were effectively used as laboratories during last century (Vandel, 1964), few remain active. In addition, the outcomes of studies on stygofauna in these caves is rarely compared to insights obtained from studies of surface freshwater organisms. This is partially due to the characteristic features of stygobionts: they are often difficult to sample in deep subterranean environments and springs can be seen as useful sampling points for them, using a combination of different methods depending on the spring morphology and hydrology (Malard et al., 2002). Particularly in springs are recommended the sampling of drifting stygobionts during spates, samplings of the spring benthic layer, and bed sediments and also the use of artificial substrates and baited traps (Malard et al., 2002).

Moreover, due to their long life cycles and low fertility, stygobionts are difficult to raise in an experimental settings. Would including springs and other surface/underground border habitats in studies on subterranean biology increase the understanding of how the constraints of the hypogean environment affect the phenotypic responses and genetic makeup of stygobionts? The rationale of this paper takes origin from this question and is to suggest that a substantial inclusion of springs (and other border habitats between underground and surface) in studies on subterranean biology, can increase the understanding of principles governing exploitation and adaptation to hypogean environments.

In this paper, we investigate the perspectives of modern researchers on considering springs not only as simple sampling points, but also as core stygobiont habitats that can broaden the insights obtained from biological studies of underground environments. We specifically performed a systematic review of the recent scientific literature to understand (i) the relevance of previous observations of typical stygobionts in springs; (ii) if these observations vary according with study discipline and the year of publication; and (iii) if these observations are more common than observations of epigean animals (i.e. aquatic surface species) in caves. By demonstrating that typical stygofauna are observed in springs more commonly than usually thought, we propose that, at least in some cases, the exploitation of border habitats be considered a non-negligible aspect of stygofauna ecology.

Survey methodology

To avoid bias, many scientific fields have largely started to favour the use of systematic evidence reviews (Acreman et al., 2020). We therefore performed a systematic review to find focused data that addressed our three aims (Table 1). For this review, we followed the Preferred Reporting Items for Systematic Reviews (PRISMA) guidelines (Page & Moher, 2017), and we searched the Web of Science database for peer-reviewed papers on both stygofauna and fauna living in spring habitats (Fig. 2). The Web of Science database contains metadata for peer-reviewed scientific articles published since 1965. We used two search strings designed to find all articles in the database that might contain observations of fauna in both caves and springs. Our search was conducted in May 2020 from Milano, Italy, using the keywords “groundwater fauna” (GF) and “spring fauna freshwater” (SFF) and searching them by topics. For the search we used a ASUS K501 PC and the Google Chrome browser, after having emptied its cache box.

Table 1 Search terms and inclusion/exclusion criteria used to describe published evidence of stygobionts in springs and to answer specific questions.

Categories	Restrictions applied	
Number of species mentioned	If clearly stated for all taxa considered in the study	
Stygofauna found in springs	If clearly stated that the species found in springs are stygobites	
Number of stygofaunal species in springs	If the number of stygobite species found in springs is clearly stated for all taxa considered in the study	
Surface fauna found underground	If clearly stated that the species found underground are of epigean origin	
Number of surface species found underground	If the number of epigean species found underground is clearly stated for all taxa considered in the study	
Species found both in caves and springs	If clearly stated that the species found both in caves and springs are epigean or stygobites	
Number of species in both (caves and springs)	If the number of stygobites or epigean species found in both is clearly stated for all taxa	
Ecology	Yes/no, depending on whether the paper provides original ecological information
(habitat of occurrence, environmental drivers etc..)	
Taxonomy	Yes/no, depending on whether the paper provides original taxonomic data	
Behavior	Yes/no, depending on whether the paper tests/reports original behavioral information/observations	
Conservation	Yes/no, depending on whether the paper explores original conservation/restoration problems or actions	
Faunal assessment	Yes/no, depending on whether the paper is mainly devoted to assess faunal composition of spring/groundwater habitat	

Figure 2 PRISMA 2020 flow diagram for the systematic reviews which included search of Web of Science database only.

One of us (BP) initially screened the articles that met our search criteria by discarding articles that were not clearly related to our study aims. She rejected articles about botany, palaeontology, geology, and all their associated subdisciplines (paleoecology, stratigraphy, geomorphology, etc.), as well as articles about subterranean environments or groundwater that did not mention animals. The articles found using the key words “spring fauna freshwater” were more difficult to screen; for the most part, the authors of these articles did not specify if their study species were part of stygofauna or not. She therefore discarded these articles only if they were not related to our study aims (e.g., papers about estuaries, palaeontology, or related topics) or if the authors provided clear evidence that the study species were not cave-dwellers.

She additionally discarded several articles that dealt strictly with agricultural sciences, biogeochemical cycles, the impacts of various pollutants (crude oils, perchlorate, etc.) on groundwater, or other environments strongly connected to groundwater (i.e., all surface water environments), but did not mention the finding of stygofauna or epigean fauna. She discarded also articles that addressed single species or taxa that are not stygofauna or typical spring fauna and have no hypogean representatives (e.g., Rechulicz, 2011 treated Pseudorasbora parva and Vilenica et al., 2016 treated mayflies). Articles concerning terrestrial environments, estuaries, swamps, mangroves, streams, rivers, lakes, and all saltwater environments, were similarly discarded (see Table 1 for more detailed information on the article selection procedure). After this first screening, she performed a second selection procedure in which she removed any articles that were unavailable or were written in a language other than English.

From the papers she collected the information listed in Table 1, including the typology of the study, distinguishing between ecology, taxonomy behaviour, conservation and fauna assessment and considering that the same paper could belong to multiple categories. Moreover, she assessed the country and the continent of origin of the data provided by the papers.

Statistical analyses

To assess the relationships between features of the selected documents and the occurrence of stygofauna in springs, we built a series of generalized linear models (GLMs) with binomial error distributions.

First, we assessed if the fact that a paper reported the occurrence of stygobionts in springs, the occurrence of surface fauna in groundwaters, or the contemporary occurrence of a stygobiont in both groundwaters and springs, was related to the paper’s field of study. Reported occurrences were used as the dependent variable, and the study disciplines (ecology, taxonomy, faunal assessment, and conservation) were used as fixed factors (Eq. (1)). We similarly built GLMs with the same dependent variable but with publication year and the search term as independent variables (Eq. (2)).

StygofaunafoundinspringsORSurfacefaunaundergroundOR

(1) OccurrenceofthesametaxainSametaxabothinspringsandgroundwaters∼Ecology+Taxonomy+Fauna.assessment+Conservation+Behaviour,family=binomial.

StygofaunafoundinspringsORSurfacefaunaundergroundOR

OccurrenceofthesametaxainSametaxabothinspringsandgroundwaters

(2) ∼Yearofpublication+as.factor(keyresearchterm),family=binomial.

Second, using only the papers selected with the GF search term, we built GLMs to test two distinct hypotheses: (1) that the number of mentions of stygobiont species in springs is higher than the number of mentions of surface species in groundwater; and (2) that missing information is different among the two situations. Both hypotheses were tested only for papers selected with the keyword “groundwater fauna” to avoid biases associated with the fact that studies found using the keyword “spring fauna freshwater” may not have sampled underground habitats. For the first test (Eq. (3)), we used the number of species mentioned by each paper as a dependent variable, including both stygofauna found in springs and surface fauna found underground. The type of observation (stygofauna in springs vs. surface fauna in groundwater) was used as a fixed factor.

(3) Numberofspeciesmentioned∼Typologyofobservation,family=nbinom2.

For the second test (Eq. (4)), we defined the dependent variable as whether it was possible to assess the number of species mentioned in a study, including both types of observations (stygofauna in springs and surface fauna in groundwater). The type of observation was used as a fixed factor, as before.

(4) NAnsoccurrence∼Typologyofobservation,family=nbinom2.

Third, we also built a series of GLMs to test if the fact that a paper reported the occurrence of stygobionts in springs, the occurrence of surface fauna in groundwaters, or the contemporary occurrence of a stygobiont in both groundwaters and springs, was related to the continent of sampling of the springs.

For all models, we checked models assumptions by verifying the absence of multicollinearity issues though VIF calculation and plotting residuals versus fitted values, versus each covariate; to avoid overdispersion bias in the models used for the second step of analysis, we built both models using a type 2 negative binomial error distribution in the package glmmTMB (Brooks et al., 2017).

We used a likelihood ratio test to assess the significance of all the fixed factors included in each GLM (Bolker et al., 2008). All analyses were performed in the R 3.6.3 environment.

Results

We retrieved 824 potentially relevant papers after removing duplicate articles. After removing articles based on the first selection criteria described above, there were 415 potentially relevant documents. After the second selection procedure, we obtained 342 papers: 275 derived from the search term “groundwater fauna” (GF) and 67 from the search term “spring fauna freshwater” (SFF). Many papers found using the “groundwater fauna” search did not specify the sampling site for the taxa considered, and many papers found using the “spring fauna freshwater” search did not clearly identify if they sampled stygofauna or not. Overall, 57 papers (representing 19% of the papers with information on sampling habitat) reported the occurrence of stygofauna in springs, 37 (11.7%) reported the occurrence of typical surface fauna underground, and 33 (11%) reported the same taxa in both springs and groundwater (Table 2). With respect to our search terms, 20% of papers dealing with GF and 16% of papers dealing with SFF described the occurrence of stygobionts in spring habitats.

Table 2 Number of papers reporting observations of stygofauna in springs, of surface fauna in groundwaters, and of the same taxa in both environments. Papers are divided based on the key words used for the systematic review: GF, groundwater fauna; SFF, spring freshwater fauna.

		Total	GF	SFF	
Stygofauna in springs	YES	57	49	8	
NO	235	195	40	
Information missing	50	31	19	
Surface fauna underground	YES	37	34	3	
NO	278	220	58	
Information missing	27	21	6	
Same taxa both in springs and groundwaters	YES	33	30	3	
NO	266	214	52	
Information missing	43	31	12	

There were 45,375 mentions of species across all papers we retrieved. Of these, 138 were stygobionts observed/sampled in springs and 46 were surface species observed in subterranean habitats.

The study disciplines covered by the papers were mainly ecology (196 papers) and faunistic assessments (177 papers). Four papers were behavioural studies, and 24 papers addressed conservation issues. There were 194 papers that encompassed multiple ad hoc definitions that we established in the methods.

Our first analysis revealed that faunal assessments are significantly more likely to report the occurrence of surface fauna in groundwater, whereas taxonomic studies are more likely to report the occurrence of the same taxon in both environments (Table 3). We did not detect any relationship between the discipline of a paper (ecology, taxonomy, etc.) and the reported occurrence of stygofauna in springs. However, we did detect a positive relationship between the year of publication and the reports of stygofauna in springs (χ2 = 4.53, P = 0.03). Papers selected using the SFF search term were significantly less likely to report the occurrence of surface taxa in groundwaters (χ2 = 4.09, P = 0.04).

Table 3 The reported occurrence of stygofauna in springs, of surface fauna in groundwaters, and of the same taxa in both habitats shown as a function of study discipline.

Relationships were assessed using generalized linear models (GLMs) followed by a likelihood ratio test. Significant relationships are reported in bold.

	Research discipline	Estimate	SE	χ2	P	
Stygofauna in springs	Ecology	0.27	0.34	0.64	0.42	
Taxonomy	0.75	0.39	3.67	0.06	
Faunal assessment	0.24	0.35	0.48	0.49	
Conservation	0.67	0.53	1.50	0.22	
Behavior	−13.84	834.76	0.98	0.32	
Surface fauna underground	Ecology	0.02	0.41	0.00	0.96	
Taxonomy	0.27	0.50	0.29	0.59	
Faunal assessment	1.09	0.45	6.62	0.01	
Conservation	0.51	0.59	0.68	0.41	
Behavior	1.30	1.20	0.94	0.33	
Same taxa both in springs and groundwaters	Ecology	0.09	0.43	0.04	0.84	
Taxonomy	0.91	0.45	3.96	0.04	
Faunal assessment	0.06	0.44	0.02	0.89	
Conservation	0.77	0.60	1.45	0.23	
Behavior	−13.15	839.90	0.51	0.48	

Most data of the papers came from Europe (59.76%) and Oceania (27.9 %); we detected no significant relationships between the continent of sampling and the occurrence of stygobionts in springs, the occurrence of surface fauna in groundwaters, and the contemporary occurrence of a stygobiont in both groundwaters and springs.

GLMs performed on papers selected using the GF search term revealed that the number of mentions of stygobiont species in springs is higher than the number of mentions of surface fauna underground (χ2 = 4.19, P = 0.04). However, there is also less information available on whether stygofauna have been observed in springs compared to whether surface species have been recorded in groundwaters (χ2 = 14.08, P < 0.001).

Discussion and perspectives

Our systematic review revealed that there are more papers about stygofauna available on Web of Science than there are papers addressing fauna and springs. Because the word “spring” is a homograph with multiple meanings, our initial search retrieved many papers that were ultimately discarded because they did not discuss fauna and spring habitats. Preliminary literature searches performed using synonyms of “spring” and/or terms that define specific spring habitats, such as “sources” or “seepage,” resulted in fewer papers. Most of these papers were already included in our analysis; however, the few that were not could be used in future study with a larger set of papers. Someone could disagree as it is likely that our research missed some papers (as an example we are sure that we missed at least 15 papers on aquatic cave-dwelling salamanders and 10 papers on the taxonomy of strictly subterranean planarians) and that further keywords should have been added, for example: stygob*, ecoton*, hypogean, subterranean, etc., but they would have increased the number of papers dealing with stygofauna without significant increase in the number of papers related to spring fauna. The large difference in the number of papers obtained with the two search terms, GF and SFF, underscores the fact that the fauna of spring habitats have received much less attention not only than the inhabitants of lakes and streams/rivers, as already pointed out by previous studies (Cantonati et al., 2011), but also than stygofauna.

We limited our review to articles archived in Web of Science; this approach was more conservative because it included only relatively recent papers published in indexed, high-impact journals that perform selective peer review. An analogous review could be performed using the Google Scholar database or a more exhaustive search of grey literature in online and physical repositories. It is possible that the older, descriptive papers archived in these databases may have reported stygofauna in springs, but it is also possible that some form of bias could arise from using older literature that has not been rigorously peer reviewed. The effects of database selection should therefore be investigated in the future.

Using both GF and SFF as search terms, we found papers that mentioned the occurrence of stygofauna in springs, of typical surface fauna in groundwaters, and of the same taxa in both environments. The number of papers that reported stygofauna in springs, as well as the number of stygobiont species that were documented in springs, represented only a fraction of the total papers and documented species but were nevertheless non-negligible. This pattern was not linked to any specific field of study; though taxonomic studies were non-significantly more likely to report stygofauna in springs.

Springs have been recognized as relevant habitats for studying stygobionts since the beginning of the 19th century. Most subterranean biologists devoted at least some of their studies to spring habitats (Culver, Holsinger & Feller, 2012; Culver & Pipan, 2014; Vandel, 1920), and Albert Vandel, the founder of the “Laboratoire Souterrain de Moulis” (France), one of the most popular subterranean laboratories in the world (Botosaneanu, 1980), stated in 1920 that a systematic study of spring habitats could furnish important insights for solving some of the evolutionary questions posed by cave-dwelling animals (Vandel, 1920). However, this concept appears only in Vandel’s conclusions and is not further developed; the idea that springs are just sampling points in non-karst areas largely prevails throughout the rest of the paper (Vandel, 1920).

Stygobionts are known to colonize the mixed assemblages of organisms residing in springs via emigration and drift from groundwaters (Malard et al., 2009; Malard et al., 2002; Manenti & Barzaghi, 2021). Typical stygobionts may be more or less permanently detected outside the spring outlet, where they can exploit different microhabitats (Malard et al., 2002; Mathieu, Essafichergui & Jeannerod, 1994). This is especially true when there is a stable supply of immigrants from karst groundwater (Mathieu, Essafi & Chergui, 1999). Our results revealed that there is a positive significant correlation with the reporting of typical stygobionts in springs and the year of publication; this means that, with respect to older studies, researchers are paying more attention when reporting data on sample collection habitats, regardless of their study discipline (ecology, taxonomy, conservation, or faunal assessment). However, in the papers that we collected, the occurrence of stygobionts in springs was often reported as either an effect of the sampling method or an occasional finding. None of the papers assessed patterns in the use of springs by stygobionts. This is true also for some papers that we missed with our search but that are well known in spring literature. As an example, Rouch (1986) defined “the hemorrhage” the flow of stygofauna pushed out from aquifers during high discharge periods through springs, erroneously considering this only as a passive mechanism. Also the case of the olm (Proteus anguinus) is challenging with both mentions of cases of passive drift of individuals from groundwaters (Aljančič, 2019) and hypotheses of active exploitations of spring habitats (Bressi, Aljancic & Lapini, 1999). In more recent times, some papers were devoted to spring discharge and the passive presence of stygobionts being flushed from “conductive” or “capacitive” aquifers (Di Lorenzo et al., 2005); other large-scale ecological surveys of springs demonstrated that in mountain areas, where species richness of stygobionts is usually poor due to the effect of Quaternary Galciations, their occurrence seems low or occasional in springs (Stoch et al., 2011), suggesting that the geographical location of springs matters and could be more in depth considered in future systematic reviews dealing with springs. Springs are also being studied with recent and ‘modern’ approaches like DNA metabarcoding techniques and eDNA that can allow to detect the presence of stygobionts in springs (West et al., 2020; White et al., 2020; Yonezawa et al., 2020) and be used in the future to assess the patterns that determine this occurrence.

The occurrence of a stygobiont species, or a species that is strictly linked to a hypogean groundwater habitat for its life cycle, in an epigean spring habitat, underlines a contradiction that might reflect the human conceptual limit of understanding borders. The human perception of limits and boundaries may be biased, as humans may recognize or emphasize abrupt distinctions when they do not exist (Pirni, 2016; Sturz & Bodily, 2016). Our results demonstrate that, at least for some stygobionts, border habitats and adjacent areas are an important part of the range and biology of stygofauna, and a proper consideration of these habitats in subterranean biology studies could provide larger perspectives. For example, stygobiont populations or individuals that exploit springs more or less permanently are likely exposed to different constraints and advantages than populations or individuals that exploit deeper aquifers. Selective pressures may therefore act differently, at least for the individuals living in springs or at the interface between subterranean and epigean habitats.

For example, different species and/or populations of the genus Niphargus, which shows typical features of stygobionts including depigmentation and the absence of eyes, have the unique ability to detect light (Fišer et al., 2016). This ability has been associated with the need to distinguish the border between surface and subterranean environments and avoid risky surface habitats (Fišer et al., 2016) where UV rays may be dangerous for a depigmented animal. However, surface habitats may also be advantageous by furnishing higher trophic resources and, at night, they are not exposed to UV light. Several studies have reported Niphargus amphipods in border habitats (Fiser et al., 2007; Manenti & Pezzoli, 2019; Marković et al., 2018). Is light perception the same between individuals from borders and individuals from deeper aquifers? Are there evolutionary adaptations for exploiting not only deep subterranean habitats but also border habitats at the interface with the surface? These questions are applicable to all stygobionts that are recurrently found in springs. Considering border habitats in addition to deeper subterranean environments therefore has the potential to double the insights obtained from studies of stygobionts. These insights could be used not only to disentangle evolution from the adaptations to the selective pressures of groundwater habitats but also to characterize the physiological responses stimulated by the interaction with different environmental conditions.

Our results further demonstrate that stygofauna are reported in springs more frequently than surface fauna are reported in groundwater, in terms of both number of papers and overall numbers of species. In recent years, a growing body of literature has shown that even the occurrence of surface species in caves is often not accidental (Lunghi, Manenti & Ficetola, 2014b, 2017), a finding that has important implications for the communities of shallow subterranean habitats (Kozel et al., 2019; Lunghi, 2018; Lunghi et al., 2020; Salvidio et al., 2020; Silva, Iniesta & Ferreira, 2020). If stygofauna occur in springs and adjacent microhabitats more commonly than surface fauna occur underground, it is likely that, at least for some stygobionts, the use of the surface environment is not accidental.

Further systematic reviews and analyses of the literature on spring fauna could be performed to investigate the countries where the largest number of studies on springs were carried out, the most studied taxa and the most studied functional traits.

A parallel can be made between springs and marine caves which can further support the idea that springs are just an ecotone that should also be studied from an ecological viewpoint (Romero, in litteris). For example, there are sea fish species that enter and exit marine caves playing a significant role in those environments’ ecology. That is the case with the cardinal fish Apogon imberbis. This is a small-sized fish distributed along the eastern Atlantic coast from Morocco to the Gulf of Guinea, including the Azores. It can be found as solitary or forming schools and is common in small crevices to marine caves, where they can be found in large densities. They show no troglomorphisms, yet they play a significant role in transferring organic material to these marine caves as mysid crustaceans do (Romero, in litteris). Like bats and Dolichopoda cave crikets (Mammola & Isaia, 2018), they tend to stay in the shelters during the day and leave the caves at night, presumably for feeding (Bussotti, Guidetti & Belmonte, 2003).

The occurrence of stygobionts in springs could affect both the dynamics of boundary habitats and, at the level of the whole stygobiont population, the intrinsic traits of the species. There are several different perspectives for how a stronger conceptual inclusion of springs in subterranean research may provide additional insights on subterranean biology. First, springs may favour intraspecific variation that could be assessed by comparative experimental studies, which would benefit studies of intraspecific dynamics between boundaries and deep areas. Second, springs can inform studies of the processes that promote adaptation to and colonization of border habitats, as research on springs could be used to distinguish possible phenotypic plasticity from local adaptations. Third, given the view of springs as useful laboratories, devoting space and infrastructure at the entrance to subterranean environments could provide important experimental opportunities.

Conclusions

Even if the transitional and ecotonal role of springs is known and studied since several decades, and the term Groundwater Dependent Ecosystems applied to springs (Fattorini et al., 2020; Rohde, Froend & Howard, 2017) allows to study the connected network of surface and subterranean ecosystems following the ‘holistic’ approach suggested by Linke et al. (2019), these concepts are rarely translated in ecological and evolutionary studies dealing with groundwater animals. The results of our systematic review broadly suggest that springs and other boundaries with surface environments should be considered and investigated as part of subterranean habitats and of the biology of at least some stygobionts. Studies of groundwater environments and stygobiont biology that do not consider springs may furnish only a limited perspective on subterranean environments, because they could exclude a priori a potential source of selective pressures for groundwater-dwelling animals. The study of groundwater-adapted organisms in subterranean aquifers has the potential to reveal new insights in several scientific fields (Pipan & Culver, 2013; Reboleira et al., 2011), but the study of the boundaries of groundwater environments, such as springs, is not only equally important, but even necessary to understand the zoology ecology and evolution of groundwater fauna.

Supplemental Information

Supplemental Information 1 Raw data and reference citations.

Click here for additional data file.

Supplemental Information 2 References for raw data.

Click here for additional data file.

Supplemental Information 3 PRISMA checklist.

Click here for additional data file.

We are grateful to Prof. David Culver and Prof. Aldemaro Romero Jr. for comments and suggestions on an early draft of this manuscript. The comments of Stefano Mammola and of an anonymous reviewer improved the quality of the manuscript.

Additional Information and Declarations

Competing Interests

Author Contributions

Data Availability

The authors declare that they have no competing interests.

Raoul Manenti conceived and designed the experiments, analyzed the data, prepared figures and/or tables, authored or reviewed drafts of the paper, and approved the final draft.

Beatrice Piazza performed the experiments, analyzed the data, prepared figures and/or tables, authored or reviewed drafts of the paper, and approved the final draft.

The following information was supplied regarding data availability:

The data taken from the references selected and the complete list of the references are available in the Supplementary Files.

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
