# Peer review of "Between darkness and light: spring habitats provide new perspectives for modern researchers on groundwater biology"

_PeerJ, doi:10.7717/peerj.11711_

## Round 0.1 · original submission · Major Revisions

As you can see I received two positive (minor revisions) reviews and one rejection. I believe your manuscript is timely and relevant but would like to see all the raised comments appropriately addressed. Please pay special attention to comments made by Reviewers 1 and 2, who have published extensively on the subject.

·

Basic reporting

An interesting systematic review of the literature about obligate fauna in springs. The goal of the authors is to discuss the potential role of springs as an integrant part of the subterranean realm.

Experimental design

In general, I think that the systematic review part is methodologically sound. I have just a few suggestions/requests for clarification:

1) I don’t think it is appropriate to call this study a meta-analysis (e.g., in the abstract “In this systematic evidence review and meta-analysis ...”). My point is that the authors didn’t combine the results of multiple scientific studies, e.g. providing a weighted average from the results of the individual studies in the form of a forest plot or similar. 
Also, the aim of the GLM analysis is to assess the relationships between features of the selected documents and the occurrence of stygofauna. This is, to my view, an analysis that falls within the domain of scientometrics/bibliometrics.
Long story short, to avoid misleading a reader, I would refer to the study as a ‘systematic literature review’ throughout the text and in the abstract.

2) Since both authors performed the “screening” of literature (sensu PRISMA, see Line 143), it would be important to report a measure of inter-operator agreement (e.g. Cohen’s Kappa)-

3) In the PRISMA diagram, I would separate “Screening” and “Eligibility”.

4) The different GLM model structures are not very clear (to me) from reading the text. Could the authors please report them also as formulas to help the reader visualize the models? See, e.g., Point 5 in Zuur & Ieno 2016

Zuur, A. F., & Ieno, E. N. (2016). A protocol for conducting and presenting results of regression‐type analyses. Methods in Ecology and Evolution, 7(6), 636-645.

Validity of the findings

(-L190): Please briefly report on model validation (see point 7 in Zuur & Ieno, 2016 mentioned in the previous comment).

-(L224-254): The entire first section of the discussion is a repetition of the result. Please omit/integrate it in the result section.

-(L263-266): Can the authors support this statement with data? How many studies have been missed by excluding these keywords?

-L230: “Our results revealed that typical stygofauna have been reported in springs more frequently in recent years.”

Maybe I’m missing something—see my previous doubt on the model structure—but I don’t think the authors have tested this. To test this, one should summarize the proportion between the paper reporting stygobionts in spring and the total number of papers for each year, and model this proportion over time (proportion ~ year).

Additional comments

I found the text generally well-written, except for the discussion that has somewhat less structure (... also note that the entire result section is repeated once in the discussion). I would suggest rethinking it by discussing each of the hypotheses tested in the same order as these are reported in the results. Also, I provided below additional minor suggestions.

I hope this revision will be useful.

Yours sincerely,
Stefano Mammola

-L77: About ecotones underground, I suggest also cite the classic:

Prous, X., Ferreira, R. L., & Martins, R. P. (2004). Ecotone delimitation: Epigean–hypogean transition in cave ecosystems. Austral Ecology, 29(4), 374-382.

-L83-84 the sentence is not fully clear, I suggest rephrasing to “the etymology of the word ...”

-L88-L91: I don’t fully understand the link between the first and the second sentence (i.e. permeability of the ecotone at night; presence of stygobionts) unless the studies mentioned (Bressi et al. 1999; Fišer 2019; Manenti & Barzaghi 2020) refer to nocturnal samplings. For example, in terrestrial subterranean systems, we observed a similar pattern, but only after confronting diurnal and nocturnal occurrence patterns (https://subtbiol.pensoft.net/article/28909/).

-L94: Can the ‘use of habitat’ be considered as a ‘trait’?

-L105: Within the frame of ‘caves as laboratory’ topic, references cited are very old. This subject has been covered extensively in these works:

Mammola, S. (2019). Finding answers in the dark: caves as models in ecology fifty years after Poulson and White. Ecography, 42(7), 1331-1351.

Mammola, S., Amorim, I. R., Bichuette, M. E., Borges, P. A., Cheeptham, N., Cooper, S. J., ... & Cardoso, P. (2020). Fundamental research questions in subterranean biology. Biological Reviews, 95(6), 1855-1872.

( I know, it’s shameless self-promotion... but see an interesting read:
https://scientistseessquirrel.wordpress.com/2020/02/04/what-should-you-do-when-reviewer-2-says-cite-my-papers/ )

-L128-133: I would suggest omitting this introduction to systematic literature in the methods. Doesn’t add much to the story.

-L140-142: Could worth mentioning also the computer/and internet browser, to ensure theoretical reproducibility of the search. See:

Pozsgai, G. et al. 2020. A comparative analysis reveals irreproducibility in searches of scientific literature. BioRxiv.

-L204: is 45,375 species correct? Does it refer to unique species or 45,375 mentions of species (mostly repeated)? Please clarify

-L208: faunal assessments → faunistic?

-L209: multiple fields of study → do you mean the “Multidisciplinary” category in WoS? Or an ad hoc definition for this paper when a paper deals with, say, ecology and evolution?

-L307: there is also the classic case of proteus washed out from caves in Carniola, that could worth mentioning. See:

Aljančič G. (2019). History of research on Proteus anguinus Laurenti 1768 in Slovenia. Folia Biologica et Geologica 60, 39-69.

-L317-318: Developments in this area have been very fast, and I think could worth mentioning more recent literature, e.g.:

West, K. M., Richards, Z. T., Harvey, E. S., Susac, R., Grealy, A., & Bunce, M. (2020). Under the karst: detecting hidden subterranean assemblages using eDNA metabarcoding in the caves of Christmas Island, Australia. Scientific reports, 10(1), 1-15.

White, N. E., Guzik, M. T., Austin, A. D., Moore, G. I., Humphreys, W. F., Alexander, J., & Bunce, M. (2020). Detection of the rare Australian endemic blind cave eel (Ophisternon candidum) with environmental DNA: implications for threatened species management in subterranean environments. Hydrobiologia, 847, 3201-3211.

Yonezawa, S., Nakano, T., Nakahama, N., Tomikawa, K., & Isagi, Y. (2020). Environmental DNA reveals cryptic diversity within the subterranean amphipod genus Pseudocrangonyx Akatsuka & Komai, 1922 (Amphipoda: Crangonyctoidea: Pseudocrangonyctidae) from central Japan. The Journal of Crustacean Biology, 40(4), 479-483.

-L365-366: This happens also with terrestrial organisms (e.g. Dolichopoda). See the previous point on comparing day/night

-L392 romero → Romero

·

Basic reporting

This is a clear review of the literature on subterranean organisms in springs. I do think that the review would be improved by at least a basic review of springs, especially the types of springs, e.g. artesian, gravity, etc. This is not to broaden the scope but because I think spring type has a major impact on what orgnaisms one can find in the spring, and the reader should at least be made aware that not all springs are alike. In my detailed comments on the manuscript, I suggest some major references.

Experimental design

The study design is fine but I wonder if it would be possible to add country or continent or latitude to the information. Just as surface and subsurface biodiversity varies geographically, so too should spring biodiversity and may be a major factor.

Validity of the findings

See above

Additional comments

None of these comments should be taken to mean the study is not worthwhile as it stands, and they are offered as possible improvements

Reviewer 3 ·

Basic reporting

I deeply regret to say that the manuscript is not correctly written. It should require a profound revision of the whole text. There are several grammatical errors throughout the text, and several sentences are unclear, and inserted in parts of the manuscript where the final conceptual frame is going definitively lost. Literature references are not sufficiently provided, and some seminal works on groundwater-fed springs not mentioned at all. The title is catchy, but the Introduction section is not clearly arranged. The whole Results section is doubled in the Discussion, generating confusion to the reader. Several statements are not adequately supported in the Discussion and Conclusion sections. The hydrogeologic dynamics governing the groundwater flow discharge at the spring outlet is neglected, as well as its strong effects on the presence of stygobites at the spring "surface". Under my opinion, and based on my knowledge of stygobites, some concepts are wrong, or incorrectly given to a reader, and body of evidence of such statements not given.

Experimental design

No comment

Validity of the findings

I regret again to say that I was unable to find any positive impact or novelty into this manuscript. The results remain to me inconclusive, and predominantly based on speculation.

Additional comments

Dear Authors,
I think that much more effort is needed for a good Opinion Paper, together with a profound knowledge of the groundwater biology and ecology. Unfortunately I was unable to read a well told story.
I would suggest the AA. to better address their discussion, by providing at first robust bodies of evidence when basic concepts related to the Groundwater Ecology are razed to the ground, which are not only coming from past research, but also from very recent scientific works.
I have provided a .pdf file by given some suggestions and improvements.
Best regards

Annotated reviews are not available for download in order to protect the identity of reviewers who chose to remain anonymous.

---

## Round 0.2 · Minor Revisions

Please address the minor issues raised by Reviewer 1. Additionally, please report the test statistic from your likelihood ratio tests (Chi-squared?): In Table 3 the column LRT should be named by the test statistic. Finally, in Fig. 1 legend, please describe the main differences between Figs. 1A and 1B. The movement of stygobionts during the night is difficult to spot.

·

Basic reporting

no comment

Experimental design

no comment

Validity of the findings

no comment

Additional comments

Thanks for this revision. My main concerns on the reporting of methods have been addressed. A few additional suggestions:

L88: the the

L91: during night → at night

L96 please remove spp.

L144: change “high-impact” to “peer-reviewed” (WOS also contain papers published in rather low impact journals)

L162-167 Please consider breaking down this very long sentence

L172 please remove also.

L243: conservation concerns → conservation issues?

L296 what is a “major subterranean biologist”?

L297: please consider spelling out the name of the lab

L368-369: To me, the example of marine caves comes a bit out of the blue. Please consider rephrasing (e.g. “A parallel can be made between springs and...”)

L391: GDE is mentioned only once. No need to define the acronym

- Figure 1: Besides the change in light conditions, don’t you expect differences in predation risk between day and night? At night there can be a largely new set of nocturnal predators.

---

## Round 0.3 · accepted · Accept

I`m happy to accept your paper, congratulations on this important contribution.